# PROBABILISTIC FEDERATED NEURAL MATCHING

## ABSTRACT

In federated learning problems, data is scattered across different servers and exchanging or pooling it is often impractical or prohibited. We develop a Bayesian nonparametric framework for federated learning with neural networks. Each data server is assumed to train local neural network weights, which are modeled through our framework. We then develop an inference approach that allows us to synthesize a more expressive global network without additional supervision or data pooling. We then demonstrate the efficacy of our approach on federated learning problems simulated from two popular image classification datasets.

## 1 INTRODUCTION

The standard machine learning paradigm involves algorithms that learn from centralized data, possibly pooled together from multiple data sources. The computations involved may be done on a single machine or farmed out to a cluster of machines. However, in the real world, data often lives in silos and amalgamating them may be rendered prohibitively expensive by communication costs, time sensitivity, or privacy concerns. Consider, for instance, data recorded from sensors embedded in wearable devices. Such data is inherently private, can be voluminous depending on the sampling rate of the sensing modality, and may be time sensitive depending on the analysis of interest. Pooling data from many users is technically challenging owing to the severe computational burden of moving large amounts of data, and fraught with privacy concerns stemming from potential data breaches that may expose the user's protected health information (PHI).

Federated learning avoids these pitfalls by obviating the need for centralized data and instead designs algorithms that learn from sequestered data sources with different data distributions. To be effective, such algorithms must be able to extract and distill important statistical patterns from various independent local learners coherently into an effective global model without centralizing data. This will allow us to avoid the prohibitively expensive cost of data communication. To achieve this, we develop and investigate a probabilistic federated learning framework with a particular emphasis on training and aggregating neural network models on siloed data.

We proceed by training local models for each data source, *in parallel*. We then match the estimated local model parameters (groups of weight vectors in the case of neural networks) across data sources to construct a global network. The matching, to be formally defined later, is governed by the posterior of a Beta-Bernoulli process (BBP) (Thibaux & Jordan, 2007; Yurochkin et al., 2018), a Bayesian nonparametric model that allows the local parameters to either match existing global ones or create a new global parameter if existing ones are poor matches. Our construction allows the size of the global network to flexibly grow or shrink as needed to best explain the observed data. Crucially, we make no assumptions about how the data is distributed between the different sources or even about the local learning algorithms. These may be adapted as necessary, for instance to account for non-identically distributed data. Further, we only require communication after the local algorithms have converged. This is in contrast with popular distributed training algorithms (Dean et al., 2012) that rely on frequent communication between the local machines. Our construction also leads to compressed global models with fewer parameters than the set of all local parameters. Unlike naive ensembles of local models, this allows us to store fewer parameters and leads to more efficient inference at test time, requiring only a single forward pass through the compressed model as opposed to $J$ forward passes, once for each local model. While techniques such as distillation (Hinton et al., 2015) allow for the cost of multiple forward passes to be amortized, training the distilled model itself requires access to data pooled across all sources, a luxury unavailable in our federated learning scenario. In summary, the key question we seek to answer in this paper is the following: given

pre-trained neural networks trained locally on non-centralized data, can we learn a compressed federated model without accessing the original data, while improving on the performance of the local networks?

The remainder of the paper is organized as follows. We briefly introduce the Beta-Bernoulli process in Section 2 before describing our model for federated learning in Section 3. We thoroughly vet the proposed models and demonstrate the utility of the proposed approach in Section 4. Finally, Section 5 discusses limitations and open questions.

## 2    BACKGROUND AND RELATED WORK

Our approach builds on tools from Bayesian nonparametrics, in particular the Beta-Bernoulli Process (BBP) (Thibaux & Jordan, 2007) and the closely related Indian Buffet Process (IBP) (Griffiths & Ghahramani, 2011). We briefly review these ideas before describing our approach. Consider a random measure $Q$ drawn from a Beta Process with mass parameter $\gamma_0$ and base measure $H$, $Q|\gamma_0, H \sim \mathrm{BP}(1, \gamma_0 H)$. It follows that $Q$ is a discrete (*not* probability) measure $Q = \sum_i \mathsf{q}_i \delta_{\theta_i}$ formed by pairs $(\mathsf{q}_i, \theta_i) \in [0,1] \times \Omega$ of weights and atoms. The weights $\{\mathsf{q}_i\}_{i=1}^\infty$ follow a stick-breaking construction (Teh et al., 2007): $\mathsf{c}_i \sim \mathrm{Beta}(\gamma_0, 1)$, $\mathsf{q}_i = \prod_{j=1}^i \mathsf{c}_j$ and the atoms are drawn i.i.d from the (scaled) base measure $\theta_i \sim H/H(\Omega)$ with domain $\Omega$. In this paper, $\Omega$ is simply $\mathbb{R}^D$ for some $D$. Subsets of atoms in the random measure $Q$ are then selected using a Bernoulli process with a base measure $Q$, $\mathcal{T}_j|Q \sim \mathrm{BeP}(Q)$ for $j = 1, \dots, J$. Each $\mathcal{T}_j$ is also a discrete measure formed by pairs $(\mathsf{b}_{ji}, \theta_i) \in \{0,1\} \times \Omega$, $\mathcal{T}_j := \sum_i \mathsf{b}_{ji} \delta_{\theta_i}$, where $\mathsf{b}_{ji}|\mathsf{q}_i \sim \mathrm{Bernoulli}(\mathsf{q}_i)\,\forall i$. Together, this hierarchical construction describes the Beta-Bernoulli process. Marginalizing $Q$ induces dependencies among $\mathcal{T}_j$, i.e. $\mathcal{T}_J|\mathcal{T}_1, \dots, \mathcal{T}_{J-1} \sim \mathrm{BeP}\left(H\frac{\gamma_0}{J} + \sum_i \frac{m_i}{J}\delta_{\theta_i}\right)$, where $m_i = \sum_{j=1}^{J-1} \mathsf{b}_{ji}$ (dependency on $J$ is suppressed in the notation for simplicity) and is sometimes called the Indian Buffet Process. The IBP can be equivalently described by the following culinary metaphor. $J$ customers arrive sequentially at a buffet and choose dishes to sample as follows, the first customer tries $\mathrm{Poisson}(\gamma_0)$ dishes. Every subsequent $j$-th customer tries each of the previously selected dishes according to their popularity, i.e. dish $i$ with probability $m_i/j$, and then tries $\mathrm{Poisson}(\gamma_0/j)$ new dishes.

The IBP, which specifies a distribution over sparse binary matrices with infinitely many columns, was originally demonstrated for latent factor analysis (Ghahramani & Griffiths, 2005). Several extensions to the IBP (and the equivalent BBP) have been developed, see Griffiths & Ghahramani (2011) for a review. Our work is related to a recent application of these ideas to distributed topic modeling (Yurochkin et al., 2018), where the authors use the BBP for modeling topics learned from multiple collections of document, and provide an inference scheme based on the Hungarian algorithm (Kuhn, 1955). Extending these ideas to federated learning of neural networks requires significant innovations and is the primary focus of our paper.

Federated learning has recently garnered attention from the machine learning community. Smith et al. (2017) pose federated learning as a multi-task learning problem, which exploits the convexity and decomposability of the cost function of the underlying support vector machine (SVM) model for distributed learning. This approach however does not extend to the neural network structure considered in our work. Others (McMahan et al., 2017) use strategies based on simple averaging of the local learner weights to learn the federated model. However, as pointed out by the authors, such naive averaging of model parameters can be disastrous for non-convex cost functions. To cope, they have to use a heuristic scheme where the local learners are forced to share the same random initialization. In contrast, our proposed framework is naturally immune to such issues since its development assumes nothing specific about how the local models were trained. Moreover, unlike the previous work of McMahan et al. (2017), our framework is non-parametric in nature and it therefore allows the federated model to flexibly grow or shrink its complexity (i.e., its sizes) to account for the varying data complexity.

There is also significant work on distributed deep learning Lian et al. (2015; 2017); Moritz et al. (2015); Li et al. (2014); Dean et al. (2012). However, the emphasis of these works is on scalable training from large data and they typically require frequent communication between the distributed nodes to be effective. Yet others explore distributed optimization with a specific emphasis on communication efficiency (Zhang et al., 2013; Shamir et al., 2014; Yang, 2013; Ma et al., 2015; Zhang & Lin, 2015). However, as pointed out by McMahan et al. (2017), these works primarily focus on

settings with convex cost functions and often assume that each distributed data source contains an equal number of data instances. These assumptions, in general, do not hold in our scenario.

## 3 PROBABILISTIC FEDERATED NEURAL MATCHING

We now apply this Bayesian nonparametric machinery to the problem of federated learning with neural networks. Our goal will be to identify subsets of neurons in each of the $J$ local models that match to neurons in other local models, and then use these to form an aggregate model where the matched parts of each of the local models are fused together.

Our approach to federated learning builds upon the following basic problem. Suppose we have trained $J$ Multilayer Perceptrons (MLPs) with one hidden layer each. For the $j$th MLP $j = 1, \ldots, J$, let $V_j^{(0)} \in \mathbb{R}^{D \times L_j}$ and $\tilde{v}_j^{(0)} \in \mathbb{R}^{L_j}$ be weights and biases of the hidden layer; $V_j^{(1)} \in \mathbb{R}^{L_j \times K}$ and $\tilde{v}_j^{(1)} \in \mathbb{R}^K$ be weights and biases of the softmax layer; $D$ be the data dimension, $L_j$ the number of neurons on the hidden layer; and $K$ the number of classes. We consider a simple architecture: $f_j(x) = \text{softmax}(\sigma(xV_j^{(0)} + \tilde{v}_j^{(0)})V_j^{(1)} + \tilde{v}_j^{(1)})$ where $\sigma(\cdot)$ is some nonlinearity (sigmoid, ReLU, etc.). Given the collection of weights and biases $\{V_j^{(0)}, \tilde{v}_j^{(0)}, V_j^{(1)}, \tilde{v}_j^{(1)}\}_{j=1}^J$ we want to learn a global neural network with weights and biases $\Theta^{(0)} \in \mathbb{R}^{D \times L}, \tilde{\theta}^{(0)} \in \mathbb{R}^L, \Theta^{(1)} \in \mathbb{R}^{L \times K}, \tilde{\theta}^{(1)} \in \mathbb{R}^K$, where $L \ll \sum_{j=1}^J L_j$ is an unknown number of hidden units of the global network to be inferred.

Our first observation is that ordering of neurons of the hidden layer of an MLP is permutation invariant. Consider any permutation $\tau(1, \ldots, L_j)$ of the $j$-th MLP – reordering columns of $V_j^{(0)}$, biases $\tilde{v}_j^{(0)}$ and rows of $V_j^{(1)}$ according to $\tau(1, \ldots, L_j)$ will not affect the outputs $f_j(x)$ for any value of $x$. Therefore, instead of treating weights as matrices and biases as vectors we view them as unordered collections of vectors $V_j^{(0)} = \{v_{jl}^{(0)} \in \mathbb{R}^D\}_{l=1}^{L_j}, V_j^{(1)} = \{v_{jl}^{(1)} \in \mathbb{R}^{L_j}\}_{l=1}^K$ and scalars $\tilde{v}_j^{(0)} = \{\tilde{v}_{jl}^{(0)} \in \mathbb{R}\}_{l=1}^{L_j}$ correspondingly.

Hidden layers in neural networks are commonly viewed as feature extractors. This perspective can be justified by the fact that last layer of a neural networks is simply a softmax regression. Since neural networks greatly outperform basic softmax regression in a majority of applications, neural networks must be supplying high quality features constructed from the input features. Mathematically, in our problem setup, every hidden neuron of $j$-th MLP represents a new feature $\tilde{x}_l(v_{jl}^{(0)}, \tilde{v}_{jl}^{(0)}) = \sigma(\langle x, v_{jl}^{(0)} \rangle + \tilde{v}_{jl}^{(0)})$. Our second observation is that each of the $(v_{jl}^{(0)}, \tilde{v}_{jl}^{(0)})$ acts as a parameterization of the corresponding neuron's feature extractor. Since each of the given MLPs was trained on the same general type of data (not necessarily homogeneous), we assume that they should share at least some feature extractors that serve the same purpose. However, due to the permutation invariance described previously, a feature extractor indexed by $l$ from the $j$-th MLP is unlikely to correspond to a feature extractor with the same index from a different MLP. In order to construct a set of global feature extractors (neurons) $\{\theta_i^{(0)} \in \mathbb{R}^D, \tilde{\theta}_i^{(0)} \in \mathbb{R}\}_{i=1}^L$ we must model the process of grouping and combining feature extractors of collection of MLPs.

### 3.1 SINGLE LAYER NEURAL MATCHING

We now present the key building block of our modeling framework, our Hierarchical BBP (Thibaux & Jordan, 2007) based model of the neurons and weights of multiple MLPs. Our generative model is as follows. First, draw a collection of global atoms (hidden layer neurons) from a Beta process prior with a base measure $H$ and mass parameter $\gamma_0$, $Q = \sum_i q_i \delta_{\theta_i}$. In our experiments we choose $H = \mathcal{N}(\boldsymbol{\mu}_0, \boldsymbol{\Sigma}_0)$ as the base measure with $\boldsymbol{\mu}_0 \in \mathbb{R}^{D+1+K}$ and diagonal $\Sigma_0$. Each $\theta_i \in \mathbb{R}^{D+1+K}$ is a concatenated vector of $[\theta_i^{(0)} \in \mathbb{R}^D, \tilde{\theta}_i^{(0)} \in \mathbb{R}, \theta_i^{(1)} \in \mathbb{R}^K]$ formed from the feature extractor weight-bias pairs with the corresponding weights of the softmax regression.

Next, for each batch (server) $j = 1, \ldots, J$, generate a batch specific distribution over global atoms (neurons):

$$Q_j | Q \sim \text{BP}(1, \gamma_j Q), \text{ then } Q_j := \sum_i p_{ji} \delta_{\theta_i}, \tag{1}$$

where the $\mathrm{p}_{ji}$s vary around corresponding $\mathrm{q}_i$. The distributional properties of $\mathrm{p}_{ji}$ are described in Thibaux & Jordan (2007). Now, for each $j = 1, \ldots, J$ select a subset of the global atoms for batch $j$ via the Bernoulli process:

$$\mathcal{T}_j := \sum_i \mathrm{b}_{ji} \delta_{\theta_i}, \text{ where } \mathrm{b}_{ji}|\mathrm{p}_{ji} \sim \text{Bern}(\mathrm{p}_{ji}) \, \forall i. \tag{2}$$

$\mathcal{T}_j$ is supported by atoms $\{\theta_i : \mathrm{b}_{ji} = 1, i = 1, 2, \ldots\}$, which represent the identities of the atoms (neurons) used by batch (server) $j$. Finally, assume that observed local atoms are noisy measurements of the corresponding global atoms:

$$\mathbf{v}_{jl}|\mathcal{T}_j \sim \mathcal{N}(\mathcal{T}_{jl}, \boldsymbol{\Sigma}_j) \text{ for } l = 1, \ldots, L_j, \text{ where } L_j := \text{card}(\mathcal{T}_j), \tag{3}$$

where $\mathbf{v}_{jl} = [v_{jl}^{(0)}, \tilde{v}_{jl}^{(0)}, v_{jl}^{(1)}]$ are the weights, biases, and softmax regression weights corresponding to the $l$-th neuron of the $j$-th MLP trained with $L_j$ neurons on the data of batch $j$.

Under this model, the key quantity to be inferred is the collection of random variables that **match** observed atoms (neurons) at any batch to the global atoms. We denote the collection of these random variables as $\{\boldsymbol{B}^j\}_{j=1}^J$, where $\boldsymbol{B}_{i,l}^j = 1$ implies that $\mathcal{T}_{jl} = \theta_i$ (there is a one-to-one correspondence between $\{\mathrm{b}_{ji}\}_{i=1}^\infty$ and $\boldsymbol{B}^j$).

**Maximum a posteriori estimation.** We now derive an algorithm for MAP estimation of global atoms for the model presented above. The objective function to be maximized is the posterior of $\{\theta_i\}_{i=1}^\infty$ and $\{\boldsymbol{B}^j\}_{j=1}^J$:

$$\underset{\{\theta_i\}, \{\boldsymbol{B}^j\}}{\arg\max} P(\{\theta_i\}, \{\boldsymbol{B}^j\}|\{\mathbf{v}_{jl}\}) \propto P(\{\mathbf{v}_{jl}\}|\{\theta_i\}, \{\boldsymbol{B}^j\}) P(\{\boldsymbol{B}^j\}) P(\{\theta_i\}). \tag{4}$$

Note that the next proposition easily follows from Gaussian-Gaussian conjugacy (Supplement 1):

**Proposition 1.** *Given $\{\boldsymbol{B}^j\}$, the MAP estimate of $\{\theta_i\}$ is given by*

$$\hat{\theta}_i = \frac{\boldsymbol{\mu}_0/\sigma_0^2 + \sum_{j,l} B_{i,l}^j \mathbf{v}_{jl}/\sigma_j^2}{1/\sigma_0^2 + \sum_{j,l} B_{i,l}^j/\sigma_j^2} \text{ for } i = 1, \ldots, L, \tag{5}$$

*where for simplicity we assume $\boldsymbol{\Sigma}_0 = \boldsymbol{I}\sigma_0^2$ and $\boldsymbol{\Sigma}_j = \boldsymbol{I}\sigma_j^2$.*

Using this fact we can cast optimization corresponding to (4) with respect to only $\{\boldsymbol{B}^j\}_{j=1}^J$. Taking natural logarithm we obtain:

$$\underset{\{\boldsymbol{B}^j\}}{\arg\max} \frac{1}{2} \sum_i \frac{\|\boldsymbol{\mu}_0/\sigma_0^2 + \sum_{j,l} B_{i,l}^j \mathbf{v}_{jl}/\sigma_j^2\|^2}{1/\sigma_0^2 + \sum_{j,l} B_{i,l}^j/\sigma_j^2} + \log(P(\{\boldsymbol{B}^j\})). \tag{6}$$

Detailed derivation of this and subsequent results are given in Supplement 1. We consider an iterative optimization approach: fixing all but one $\boldsymbol{B}^j$ we find corresponding optimal assignment, then pick a new $j$ at random and proceed until convergence. In the following we will use notation $-j$ to say "all but $j$". Let $L_{-j} = \max\{i : B_{i,l}^{-j} = 1\}$ denote number of active global weights outside of group $j$. We now rearrange the *first* term of (6) by partitioning it into $i = 1, \ldots, L_{-j}$ and $i = L_{-j} + 1, \ldots, L_{-j} + L_j$. We are interested in solving for $\boldsymbol{B}^j$, hence we can modify objective function by subtracting terms independent of $\boldsymbol{B}^j$ and noting that $\sum_l B_{i,l}^j \in \{0, 1\}$, i.e. it is 1 if some neuron from batch $j$ is matched to global neuron $i$ and 0 otherwise:

$$\sum_{i=1}^{L_{-j}+L_j} \sum_{l=1}^{L_j} B_{i,l}^j \left( \frac{\|\boldsymbol{\mu}_0/\sigma_0^2 + \mathbf{v}_{jl}/\sigma_j^2 + \sum_{-j,l} B_{i,l}^j \mathbf{v}_{jl}/\sigma_j^2\|^2}{1/\sigma_0^2 + 1/\sigma_j^2 + \sum_{-j,l} B_{i,l}^j/\sigma_j^2} - \frac{\|\boldsymbol{\mu}_0/\sigma_0^2 + \sum_{-j,l} B_{i,l}^j \mathbf{v}_{jl}/\sigma_j^2\|^2}{1/\sigma_0^2 + \sum_{-j,l} B_{i,l}^j/\sigma_j^2} \right). \tag{7}$$

Now we consider the *second* term of (6):

$$\log P(\{\boldsymbol{B}^j\}) = \log P(\boldsymbol{B}^j|\boldsymbol{B}^{-j}) + \log P(\boldsymbol{B}^{-j}).$$

First, because we are optimizing for $\boldsymbol{B}^j$, we can ignore $\log P(\boldsymbol{B}^{-j})$. Second, due to exchangeability of batches (i.e. customers of the IBP), we can always consider $\boldsymbol{B}^j$ to be the last batch (i.e. last

customer of the IBP). Let $m_i^{-j} = \sum_{-j,l} B_{i,l}^j$ denote number of times batch weights were assigned to global weight $i$ outside of group $j$. We now obtain the following:

$$\sum_{i=1}^{L_{-j}} \sum_{l=1}^{L_j} B_{i,l}^j \log \frac{m_i^{-j}}{J - m_i^{-j}} + \sum_{i=L_{-j}+1}^{L_{-j}+L_j} \sum_{l=1}^{L_j} B_{i,l}^j \left( \log \frac{\gamma_0}{J} - \log(i - L_{-j}) \right). \tag{8}$$

Combining (7) and (8) we obtain the assignment cost objective, which we solve with the Hungarian algorithm.

**Proposition 2.** *The assignment cost specification for finding $\mathbf{B}^j$ is:*

$$C_{i,l}^j = - \begin{cases} \frac{\|\boldsymbol{\mu}_0/\sigma_0^2 + \mathbf{v}_{jl}/\sigma_j^2 + \sum_{-j,l} B_{i,l}^j \mathbf{v}_{jl}/\sigma_j^2\|^2}{1/\sigma_0^2 + 1/\sigma_j^2 + \sum_{-j,l} B_{i,l}^j/\sigma_j^2} - \frac{\|\boldsymbol{\mu}_0/\sigma_0^2 + \sum_{-j,l} B_{i,l}^j \mathbf{v}_{jl}/\sigma_j^2\|^2}{1/\sigma_0^2 + \sum_{-j,l} B_{i,l}^j/\sigma_j^2} + \log \frac{m_i^{-j}}{J - m_i^{-j}}, & i \leq L_{-j} \\ \frac{\|\boldsymbol{\mu}_0/\sigma_0^2 + \mathbf{v}_{jl}/\sigma_j^2\|^2}{1/\sigma_0^2 + 1/\sigma_j^2} - \frac{\|\boldsymbol{\mu}_0/\sigma_0^2\|^2}{1/\sigma_0^2} - 2 \log(i - L_{-j}) + 2 \log \frac{\gamma_0}{J}, & L_{-j} < i \leq L_{-j} + L_j. \end{cases}$$
$$\tag{9}$$

*We then apply the Hungarian algorithm described in Supplement 1 to find the* minimizer *of* $\sum_i \sum_l B_{i,l}^j C_{i,l}^j$ *and obtain the neuron matching assignments.*

We summarize the overall single layer inference procedure in Figure 1 below.

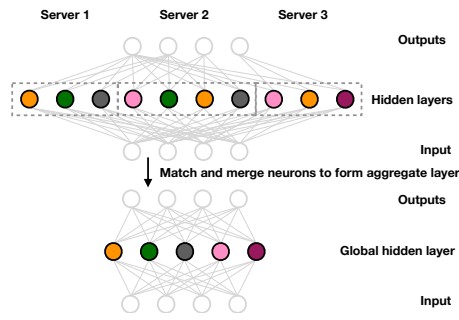

**Algorithm 1** Single Layer Neural Matching

1: Collect hidden layers from the $J$ servers and form $\mathbf{v}_{jl}$.
2: Form assignment cost matrix per (9).
3: Compute matching assignments $B^j$ using the Hungarian algorithm (Supplement 1).
4: Enumerate all resulting unique global neurons and use (5) to infer the associated global weight vectors from all instances of the global neurons across the $J$ servers.
5: Concatenate the global neurons and the inferred weights and biases to form the new global hidden layer.

Figure 1: Single layer Probabilistic Neural Matching algorithm showing matching of three MLPs. Nodes in the graphs indicate neurons, neurons of the same color have been matched. Our approach consists of using the corresponding neurons in the output layer to convert the neurons in each of the $J$ servers to weight vectors referencing the output layer. These weight vectors are then used to form a cost matrix, which the Hungarian algorithm then uses to do the matching. Finally, the matched neurons are then aggregated and averaged to form the new layer of the global model.

## 3.2 MULTILAYER NEURAL MATCHING

The model we have presented thus far can handle any arbitrary width single layer neural network, which is known to be theoretically sufficient for approximating any function of interest (Hornik et al., 1989). However, deep neural networks with moderate layer widths are known to be beneficial both practically (LeCun et al., 2015) and theoretically (Poggio et al., 2017). We extend our neural matching approach to these deep architectures by defining a generative model of deep neural network weights from outputs back to inputs (top-down). Let $C$ denote the number of hidden layers and $L^c$ the number of neurons on the $c$th layer. Then $L^{C+1} = K$ is the number of labels and $L^0 = D$ is the input dimension. In the top down approach, we consider the global atoms to be vectors of outgoing weights from a neuron instead of weights forming a neuron as it was in the single hidden layer model. This change is needed to avoid base measures with unbounded dimensions.

Starting with the top hidden layer $c = C$, we generate each layer following a model similar to that used in the single layer case. For each layer we generate a collection of global atoms and select a subset of them for each batch using Hierarchical Beta-Bernoulli process construction. $L^{c+1}$ is the number of neurons on the layer $c + 1$, which controls the dimension of the atoms in layer $c$.

**Definition 1** (Multilayer generative process). *Starting with layer $c = C$, generate (as in the single layer process)*

$$Q^c|\gamma_0^c, H^c, L^{c+1} \sim \text{BP}(1, \gamma_0^c H^c), \text{ then } Q^c = \sum_i q_i^c \delta_{\theta_i^c}, \ \theta_i^c \sim \mathcal{N}(\boldsymbol{\mu}_0^c, \boldsymbol{\Sigma}_0^c), \ \boldsymbol{\mu}_0^c \in \mathbb{R}^{L^{c+1}}$$

$$Q_j^c|\gamma_j^c, Q^c \sim \text{BP}(1, \gamma_j^c Q^c), \ \mathcal{T}_j^c := \sum_i \text{b}_{ji}^c \delta_{\theta_i^c}, \text{ where } \text{b}_{ji}^c|\text{p}_{ji}^c \sim \text{Bern}(\text{p}_{ji}^c). \tag{10}$$

*This $\mathcal{T}_j^c$ is the set of global atoms (neurons) used by batch $j$ in layer $c$, it is contains atoms $\{\theta_i^c : \text{b}_{ji}^c = 1, i = 1, 2, \ldots\}$. Finally, generate the observed local atoms:*

$$\mathbf{v}_{jl}^c|\mathcal{T}_j^c, \sim \mathcal{N}(\mathcal{T}_{jl}^c, \boldsymbol{\Sigma}_j^c) \text{ for } l = 1, \ldots, L_j^c, \text{ where } L_j^c := \text{card}(\mathcal{T}_j^c). \tag{11}$$

*Next, compute the generated number of global neurons $L^c = \text{card}\{\cup_{j=1}^J \mathcal{T}_j^c\}$ and repeat this generative process for the next layer $c - 1$. Repeat until all layers are generated ($c = C, \ldots, 1$).*

An important difference from the single layer model is that we should now set to 0 some of the dimensions of $\mathbf{v}_{jl}^c \in \mathbb{R}^{L^{c+1}}$ since they correspond to weights outgoing to neurons of the layer $c + 1$ not present on the batch $j$, i.e. $\mathbf{v}_{jli}^c := 0$ if $\text{b}_{ji}^{c+1} = 0$ for $i = 1, \ldots, L^{c+1}$. The resulting model can be understood as follows. There is a global fully connected neural network with $L^c$ neurons on layer $c$ and there are $J$ partially connected neural networks with $L_j^c$ active neurons on layer $c$, while weights corresponding to the remaining $L^c - L_j^c$ neurons are zeroes and have no effect locally.

**Remark 1.** *Our model can conceptually handle permuted ordering of the input dimensions across batches, however in most practical cases the ordering of input dimensions is consistent across batches, making the weights connecting the first hidden layer to the input only permutation invariant on the side of the first hidden layer. Similarly to how all weights were concatenated in the single hidden layer model, we consider $\boldsymbol{\mu}_0^c \in \mathbb{R}^{D+L^{c+1}}$ for $c = 1$. We also note that the bias term can be added to the model, we omitted it to simplify notation.*

**Inference** Following the top-down generative model, we adopt a greedy inference procedure that first infers the matching of the top layer and then proceeds down the layers of the network. This is possible because the generative process for each layer depends only on the identity and number of the global neurons in the layer above it, hence once we infer the $c + 1$th layer of the global model we can apply the single layer inference algorithm (Algorithm 1) to the $c$th layer. This greedy setup is illustrated in Figure 1 in Supplement 2.

The per-layer inference derivation is a straightforward copy of the single layer case, yielding the following propositions.

**Proposition 3.** *The assignment cost specification for finding $\boldsymbol{B}^{j,c}$ is:*

$$C_{i,l}^{j,c} = - \begin{cases} \frac{\|\boldsymbol{\mu}_0^c/(\sigma_0^c)^2 + \mathbf{v}_{jl}^c/(\sigma_j^c)^2 + \sum_{-j,l} B_{i,l}^{j,c} \mathbf{v}_{jl}^c/(\sigma_j^c)^2\|^2}{1/(\sigma_0^c)^2 + 1/(\sigma_j^c)^2 + \sum_{-j,l} B_{i,l}^{j,c}/(\sigma_j^c)^2} - \frac{\|\boldsymbol{\mu}_0^c/(\sigma_0^c)^2 + \sum_{-j,l} B_{i,l}^{j,c} \mathbf{v}_{jl}^c/(\sigma_j^c)^2\|^2}{1/(\sigma_0^c)^2 + \sum_{-j,l} B_{i,l}^{j,c}/(\sigma_j^c)^2} + \log \frac{m_i^{-j,c}}{J - m_i^{-j,c}}, & i \leq L_{-j}^c \\ \frac{\|\boldsymbol{\mu}_0^c/(\sigma_0^c)^2 + \mathbf{v}_{jl}^c/(\sigma_j^c)^2\|^2}{1/(\sigma_0^c)^2 + 1/(\sigma_j^c)^2} - \frac{\|\boldsymbol{\mu}_0^c/(\sigma_0^c)^2\|^2}{1/(\sigma_0^c)^2} - 2\log(i - L_{-j}^c) + 2\log \frac{\gamma_0}{J}, & L_{-j}^c < i \leq L_{-j}^c + L_j^c, \end{cases}$$

*where for simplicity we assume $\boldsymbol{\Sigma}_0^c = \boldsymbol{I}(\sigma_0^c)^2$ and $\boldsymbol{\Sigma}_j^c = \boldsymbol{I}(\sigma_j^c)^2$. We then apply the Hungarian algorithm to find the minimizer of $\sum_i \sum_l B_{i,l}^{j,c} C_{i,l}^{j,c}$ and obtain the neuron matching assignments.*

**Proposition 4.** *Given the assignment $\{\boldsymbol{B}^{j,c}\}$, the MAP estimate of $\{\theta_i^c\}$ is given by*

$$\hat{\theta}_i^c = \frac{\boldsymbol{\mu}_0^c/(\sigma_0^c)^2 + \sum_{j,l} B_{i,l}^{j,c} \mathbf{v}_{jl}^c/(\sigma_j^c)^2}{1/(\sigma_0^c)^2 + \sum_{j,l} B_{i,l}^{j,c}/(\sigma_j^c)^2} \text{ for } i = 1, \ldots, L. \tag{12}$$

We combine these propositions and summarize the overall multilayer inference procedure in Algorithm 1 in Supplement 2.

### 3.3 STREAMING NEURAL MATCHING

In this section we propose an extension of our modeling framework to handle streaming data. Such data naturally arises in many federated learning settings. Consider, again the example of wearable

devices. Data recorded by sensors on these devices is naturally temporal and memory constraints typically require streaming processing of the data.

Bayesian paradigm naturally fits into the streaming scenario - posterior of step $s$ becomes prior for step $s+1$. We generalize our single hidden layer model to streaming setting (our approach naturally extends to multilayer scenario).

The differences in the generative model effect (2) and (3), which become:

$$\mathcal{T}_j^s := \sum_i \mathbf{b}_{ji}^s \delta_{\theta_i}, \text{ where } \mathbf{b}_{ji}^s | \mathbf{p}_{ji} \sim \text{Bern}(\mathbf{p}_{ji}). \tag{13}$$

$$\mathbf{v}_{jl}^s | \mathcal{T}_j^s \sim \mathcal{N}(\mathcal{T}_{jl}^s, \boldsymbol{\Sigma}_j) \text{ for } l = 1, \ldots, L_j^s, \ s = 1, \ldots, S. \tag{14}$$

We derive cost expression for the streaming extension in the Supplementary.

## 4 EXPERIMENTS

To verify our methodology we simulate federated learning scenarios using two standard datasets: MNIST and CIFAR-10. We randomly partition each of these datasets into $J$ batches. Two partition strategies are of interest: (a) homogeneous partition when each batch has approximately equal proportion of each of the $K$ classes; and (b) heterogeneous when batch sizes and class proportions are unbalanced. We achieve the latter by simulating $\mathbf{p}_k \sim \text{Dir}_J(0.2)$ and allocating $\mathbf{p}_{k,j}$ proportion of instances of class $k$ to batch $j$. Note that due to the small concentration parameter (0.2) of the Dirichlet distribution, some sampled batches may not have any examples of certain classes of data. For each pair of partition strategy and dataset we run 10 trials to obtain mean accuracies and standard deviations. In our empirical studies below, we will show that our framework can aggregate multiple local neural networks (NNs) trained independently on different batches of data into an efficient, modest-size global neural network that performs competitively against ensemble methods and outperforms distributed optimization.

**Baselines satisfying the constraints.** First, we will conduct experiments to demonstrate that PFNM is the best performing approach among methods restricted to single communication, compressed global model and no access to data after training of the local models. Studying such constraints is not only important in the context of federated learning, but also to understand model averaging of neural networks in the parameter space. A good neural network averaging approach may serve as initialization for Knowledge Distillation when additional data is available or for distributed optimization when it is possible to perform additional communication rounds.

McMahan et al. (2017) (Figure 1 in their paper) showed that naive averaging of weights of two independently trained neural networks does not perform well, unless these weights were trained with same initial values. We will show experimentally that even with shared initialization, Federated Averaging (McMahan et al., 2017) with single post-training communication quickly degrades for more than 2 networks and/or when trained on datasets with different class distributions. On the contrary, PFNM does not require shared initialization and can produce meaningful average of many neural networks in the parameter space. We also compare to nonparametric clustering of weight vectors based on DP-means (Kulis & Jordan, 2012). This method is inspired by the Dirichlet Process mixtures (Ferguson, 1973; Antoniak, 1974), and may serve as an alternative approach to our Beta Process based construction. We note that, to the best of our knowledge, DP-means has not been considered in such context previously. Additionally, the average test set performance of the local models serves as a basic baseline.

Test set performance for varying number of batches and homogeneous and heterogeneous partitionings of MNIST are summarized in Fig. 2a and 2c. PFNM with $\gamma_0 = 1$ (hyperparameters $\sigma_0^2 = 10$ and $\sigma^2 = 1$ are fixed across experiments) consistently outperforms all baselines. We also consider a degenerate case of PFNM with $\gamma_0 = 10^{-5}$. It can be seen from Proposition 2 that $\gamma_0/J$ controls the size of the global model and when set to very small value will result in the global model of the size of the local model, however potentially at a cost of performance quality. In this experiment each of the $J$ local neural networks is trained for 10 epochs and has 100 neurons (for Federated averaging we consider 300 neurons per local network to increase global model capacity, since this approach constraints the global model size to be equal to local model) and maximum number of neurons for PFNM and DP-means is truncated at 700. Fig. 2b and 2d summarize global model sizes, which

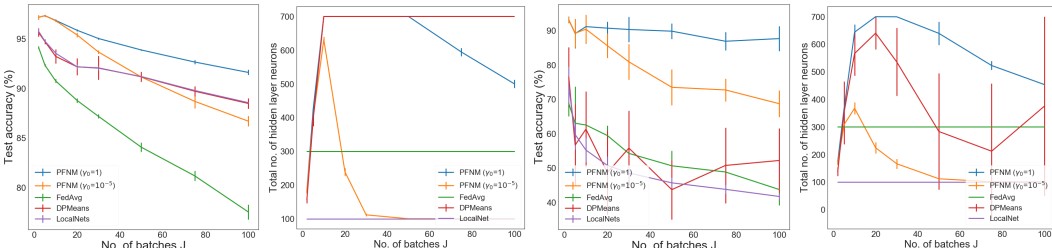

(a) MNIST homogeneous    (b) MNIST homogeneous    (c) MNIST heterogeneous    (d) MNIST heterogeneous

Figure 2: Baselines satisfying the constraints: Test accuracy and model size comparison for varying number of batches $J$ (total data size fixed, hence decreasing batch size)

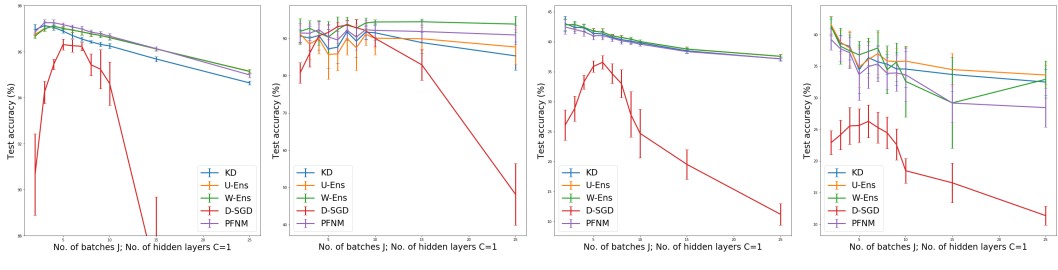

(a) MNIST homogeneous    (b) MNIST heterogeneous    (c) CIFAR homogeneous    (d) CIFAR heterogeneous

Figure 3: Comparison to baselines with extra resources: Test accuracy comparison for varying number of batches $J$ (total data size fixed, hence decreasing batch size)

show significant compression over maximum possible $100J$. Our experiments demonstrate that it is possible to efficiently perform model averaging of neural networks in the parameter space by accounting for permutation invariance of the hidden neurons. We reiterate that the global model learned by PFNM may either be used as final solution for a federated learning problem or serve as an initialization to obtain better performance with distributed optimization, Federated Averaging or Knowledge Distillation when some of our problem constraints are relaxed.

**Baselines with extra resources.** We next consider four additional baselines, however each of them violates at least one of the three constraints of our federated learning problem, i.e. no data pooling, infrequent communication, and a modest-size global model. Our goal iss to demonstrate that PFNM is competitive even when put at a disadvantage. Uniform ensemble (U-Ens) (Dietterich, 2000) is a classic technique for aggregating multiple learners. For a given test case, each batch neural network outputs class probabilities which are averaged across batches to produce the prediction of class probabilities. The disadvantage of this approach is high computational cost at testing time since it essentially stacks all batch neural networks into a master classifier with $\sum_{j,c} L_j^c$ hidden units. Weighted ensemble (W-Ens) is a heuristic extension for heterogeneous partitioning - where class $k$ probability of batch $j$ is weighted by the proportion of instances of class $k$ on batch $j$ when taking the average across batch network outputs. Knowledge distillation (KD) Hinton et al. (2015) is an extension of ensemble, where a new, modest size neural network is trained to mimic the behavior of an ensemble. This, however, requires pooling training examples on the master node. Our final baseline is the distributed optimization approach downpour SGD (D-SGD) of Dean et al. (2012). The limitation of this method is that it requires frequent communication between batch servers and the master node in order to exchange gradient information and update local copies of weights. In our experiments downpour SGD was allowed to communicate once every training epoch (total of 10 rounds of communications), while our method and other baselines only communicated once, i.e. after the batch neural networks have been trained.

We compare Probabilistic Federated Neural Matching (PFNM) against the above four extra-resource baselines for varying number of batches $J$ (Fig. 3). When number of batches grows, average size of a single batch decreases and corresponding neural networks do not converge to a good solution (or

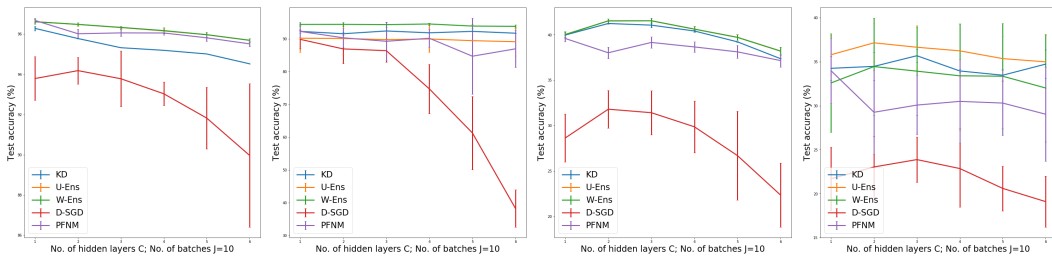

(a) MNIST homogeneous  (b) MNIST heterogeneous  (c) CIFAR homogeneous  (d) CIFAR heterogeneous

Figure 4: Comparison to baselines with extra resources: Test accuracy comparison for varying number of layers $C$ with $J = 10$

result in a bad gradient after an epoch). This significantly degrades performance of the downpour SGD and also affects PFNM in the case of heterogeneous CIFAR-10 (Fig. 3d). We observe that D-SGD at first improves with increasing number of batches and then drops down in performance abruptly — at first increasing number of batches essentially increases number of communications, since each batch sends gradients to the server, without hurting the quality of gradients, however when size of batches decreases gradients become worse and D-SGD behaves poorly. On the other hand, ensemble approaches only require a collection of weak classifiers to perform well, hence their performance does not noticeably degrade as the quality of batch neural networks deteriorates. This advantage comes at a price of high computational burden when making a prediction, since we need to do a forward pass for an input observation through each of the batch networks. Interestingly, weighted ensemble performs worse than uniform ensemble on heterogeneous CIFAR-10 case - this again could be due to the low quality of batch networks which hurts our method and makes uniform ensemble more robust than weighted. In the second experiment we fix $J = 10$ and consider multi-layer batch neural networks with number of layers $C$ from 1 to 6. We see (Fig. 4) that our multilayer PFNM can handle deep networks as it continues to be comparable to ensemble techniques and out-perform D-SGD. In the Supplementary we analyze sizes of the master neural network learned by PFNM, parameter sensitivity, streaming extension and explore performance of downpour SGD with more frequent communications. We conclude that for federated learning applications when prediction time is limited (hence ensemble approaches are not suitable) and communication is expensive, PFNM is a strong solution candidate.

## 5 DISCUSSION

In this work we have developed models for matching fully connected networks, and experimentally demonstrated the capabilities of our methodology, particularly when prediction time is limited and communication is expensive. We also observed the importance of convergent local neural networks that serve as inputs to our matching algorithms. Poor quality local neural network weights will affect the quality of the master network. In future work we plan to explore more sophisticated ways to account for uncertainty in the weights of small batches. Additionally, our matching approach is completely unsupervised – incorporating some form of supervised signal may help to improve the performance of the global network when local networks are low quality. Finally, it is of interest to extend our modeling framework to other architectures such as Convolutional Neural Networks (CNNs) and Recurrent Neural Networks (RNNs). The permutation invariance necessitating matching inference arises in CNNs too — any permutation of the filters results in same output, however additional bookkeeping is needed due to pooling operations.

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

# SUPPLEMENTARY MATERIAL FOR PROBABILISTIC FEDERATED NEURAL MATCHING

**Anonymous authors**

## 1  SINGLE HIDDEN LAYER INFERENCE

The goal of maximum a posteriori (MAP) estimation is to maximize posterior probability of the latent variables: global atoms $\{\theta_i\}_{i=1}^{\infty}$ and assignments of observed neural network weight estimates to global atoms $\{\boldsymbol{B}^j\}_{j=1}^{J}$, given estimates of the batch weights $\{\mathbf{v}_{jl}$ for $l = 1, \dots, L_j\}_{j=1}^{J}$:

$$\arg\max_{\{\theta_i\},\{\boldsymbol{B}_j\}} P(\{\theta_i\}, \{\boldsymbol{B}^j\}|\{\mathbf{v}_{jl}\}) \propto P(\{\mathbf{v}_{jl}\}|\{\theta_i\}, \{\boldsymbol{B}^j\})P(\{\boldsymbol{B}^j\})P(\{\theta_i\}). \tag{1}$$

**MAP estimates given matching (Proposition 1 in the main text)**   First we note that given $\{\boldsymbol{B}^j\}$ it is straightforward to find MAP estimates of $\{\theta_i\}$ based on Gaussian-Gaussian conjugacy:

$$\hat{\theta}_i = \frac{\sum_{j,l} B_{i,l}^j \mathbf{v}_{jl}/\sigma_j^2}{1/\sigma_0^2 + \sum_{j,l} B_{i,l}^j/\sigma_j^2} \text{ for } i = 1, \dots, L, \tag{2}$$

where $L = \max\{i : B_{i,l}^j = 1 \text{ for } l = 1, \dots, L_j, \ j = 1, \dots, J\}$ is the number of active global atoms, which is an (unknown) latent random variable identified by $\{\boldsymbol{B}^j\}$. For simplicity we assume $\boldsymbol{\Sigma}_0 = \boldsymbol{I}\sigma_0^2$, $\boldsymbol{\Sigma}_j = \boldsymbol{I}\sigma_j^2$ and $\boldsymbol{\mu}_0 = 0$.

**Inference of atom assignment.**   We can now cast optimization corresponding to (1) with respect to only $\{\boldsymbol{B}^j\}_{j=1}^{J}$. Taking natural logarithm we obtain:

$$-\frac{1}{2} \sum_i \left( \frac{\|\hat{\theta}_i\|^2}{\sigma_0^2} + D \log(2\pi\sigma_0^2) + \sum_{j,l} B_{i,l}^j \frac{\|\mathbf{v}_{jl} - \hat{\theta}_i\|^2}{\sigma_j^2} \right) + \log(P(\{\boldsymbol{B}^j\})). \tag{3}$$

Let us first simplify the first term of (3):

$$-\frac{1}{2} \sum_i \left( \frac{\|\hat{\theta}_i\|^2}{\sigma_0^2} + D \log(2\pi\sigma_0^2) + \sum_{j,l} B_{i,l}^j \frac{\|\mathbf{v}_{jl} - \hat{\theta}_i\|^2}{\sigma_j^2} \right)$$

$$= -\frac{1}{2} \sum_i \left( \frac{\langle \hat{\theta}_i, \hat{\theta}_i \rangle}{\sigma_0^2} + D \log(2\pi\sigma_0^2) + \sum_{j,l} B_{i,l}^j \frac{\langle \mathbf{v}_{jl}, \mathbf{v}_{jl} \rangle - 2\langle \mathbf{v}_{jl}, \hat{\theta}_i \rangle + \langle \hat{\theta}_i, \hat{\theta}_i \rangle}{\sigma_{jl}^2} \right)$$

$$\cong -\frac{1}{2} \sum_i \left( \langle \hat{\theta}_i, \hat{\theta}_i \rangle \left( \frac{1}{\sigma_0^2} + \sum_{j,l} \frac{B_{i,l}^j}{\sigma_j^2} \right) + D \log(2\pi\sigma_0^2) - 2\langle \hat{\theta}_i, \sum_{j,l} B_{i,l}^j \frac{\mathbf{v}_{jl}}{\sigma_j^2} \rangle \right) \tag{4}$$

$$= \frac{1}{2} \sum_i \left( \langle \hat{\theta}_i, \hat{\theta}_i \rangle \left( \frac{1}{\sigma_0^2} + \sum_{j,l} \frac{B_{i,l}^j}{\sigma_j^2} \right) - D \log(2\pi\sigma_0^2) \right)$$

$$= \frac{1}{2} \sum_i \left( \frac{\|\sum_{j,l} B_{i,l}^j \mathbf{v}_{jl}/\sigma_j^2\|^2}{1/\sigma_0^2 + \sum_{j,l} B_{i,l}^j/\sigma_j^2} - D \log(2\pi\sigma_0^2) \right).$$

We consider an iterative optimization approach: fixing all but one $\boldsymbol{B}^j$ we find corresponding optimal assignment, then pick a new $j$ at random and proceed until convergence. In the following we will

use notation $-j$ to say "all but $j$". Let $L_{-j} = \max\{i : B_{i,l}^{-j} = 1\}$ denote number of active global weights outside of group $j$. We now rearrange (4) by partitioning it into $i = 1, \ldots, L_{-j}$ and $i = L_{-j}+1, \ldots, L_{-j}+L_j$. We are interested in solving for $\boldsymbol{B}^j$, hence we can modify objective function by subtracting terms independent of $\boldsymbol{B}^j$:

$$\sum_i \left( \frac{\|\sum_{j,l} B_{i,l}^j \mathbf{v}_{jl}/\sigma_j^2\|^2}{1/\sigma_0^2 + \sum_{j,l} B_{i,l}^j/\sigma_j^2} - D\log(2\pi\sigma_0^2) \right)$$

$$\cong \sum_{i=1}^{L_{-j}} \left( \frac{\|\sum_l B_{i,l}^j \mathbf{v}_{jl}/\sigma_j^2 + \sum_{-j,l} B_{i,l}^j \mathbf{v}_{jl}/\sigma_j^2\|^2}{1/\sigma_0^2 + \sum_l B_{i,l}^j/\sigma_j^2 + \sum_{-j,l} B_{i,l}^j/\sigma_j^2} - \frac{\|\sum_{-j,l} B_{i,l}^j \mathbf{v}_{jl}/\sigma_j^2\|^2}{1/\sigma_0^2 + \sum_{-j,l} B_{i,l}^j/\sigma_j^2} \right) \quad (5)$$

$$+ \sum_{i=L_{-j}+1}^{L_{-j}+L_j} \left( \frac{\|\sum_l B_{i,l}^j \mathbf{v}_{jl}/\sigma_j^2\|^2}{1/\sigma_0^2 + \sum_l B_{i,l}^j/\sigma_j^2} \right).$$

Now observe that $\sum_l B_{i,l}^j \in \{0,1\}$, i.e. it is 1 if some neuron from batch $j$ is matched to global neuron $i$ and 0 otherwise. Due to this we can rewrite (5) as a linear sum assignment problem:

$$\sum_{i=1}^{L_{-j}} \sum_{l=1}^{L_j} B_{i,l}^j \left( \frac{\|\mathbf{v}_{jl}/\sigma_j^2 + \sum_{-j,l} B_{i,l}^j \mathbf{v}_{jl}/\sigma_j^2\|^2}{1/\sigma_0^2 + 1/\sigma_j^2 + \sum_{-j,l} B_{i,l}^j/\sigma_j^2} - \frac{\|\sum_{-j,l} B_{i,l}^j \mathbf{v}_{jl}/\sigma_j^2\|^2}{1/\sigma_0^2 + \sum_{-j,l} B_{i,l}^j/\sigma_j^2} \right)$$

$$+ \sum_{i=L_{-j}+1}^{L_{-j}+L_j} \sum_{l=1}^{L_j} B_{i,l}^j \left( \frac{\|\mathbf{v}_{jl}/\sigma_j^2\|^2}{1/\sigma_0^2 + 1/\sigma_j^2} \right). \quad (6)$$

Now we consider second term of (3):

$$\log P(\{\boldsymbol{B}^j\}) = \log P(\boldsymbol{B}^j|\boldsymbol{B}^{-j}) + \log P(\boldsymbol{B}^{-j}).$$

First, because we are optimizing for $\boldsymbol{B}^j$, we can ignore $\log P(\boldsymbol{B}^{-j})$. Second, due to exchangeability of batches (i.e. customers of the IBP), we can always consider $\boldsymbol{B}^j$ to be the last batch (i.e. last customer of the IBP). Let $m_i^{-j} = \sum_{-j,l} B_{i,l}^j$ denote number of times batch weights were assigned to global atom $i$ outside of group $j$. We now obtain the following:

$$\log P(\boldsymbol{B}^j|\boldsymbol{B}^{-j}) \cong \sum_{i=1}^{L_{-j}} \left( \left( \sum_{l=1}^{L_j} B_{i,l}^j \right) \log \frac{m_i^{-j}}{J} + \left( 1 - \sum_{l=1}^{L_j} B_{i,l}^j \right) \log \frac{J - m_i^{-j}}{J} \right)$$

$$- \log \left( \sum_{i=L_{-j}+1}^{L_{-j}+L_j} \sum_{l=1}^{L_j} B_{i,l}^j \right)! + \left( \sum_{i=L_{-j}+1}^{L_{-j}+L_j} \sum_{l=1}^{L_j} B_{i,l}^j \right) \log \frac{\gamma_0}{J}. \quad (7)$$

We now rearrange (7) as linear sum assignment problem:

$$\sum_{i=1}^{L_{-j}} \sum_{l=1}^{L_j} B_{i,l}^j \log \frac{m_i^{-j}}{J - m_i^{-j}} + \sum_{i=L_{-j}+1}^{L_{-j}+L_j} \sum_{l=1}^{L_j} B_{i,l}^j \left( \log \frac{\gamma_0}{J} - \log(i - L_{-j}) \right). \quad (8)$$

Combining (6) and (8) we arrive at the cost specification for finding $\boldsymbol{B}^j$ as *minimizer* of $\sum_i \sum_l B_{i,l}^j C_{i,l}^j$, where:

$$C_{i,l}^j = - \begin{cases} \frac{\|\mathbf{v}_{jl}/\sigma_j^2 + \sum_{-j,l} B_{i,l}^j \mathbf{v}_{jl}/\sigma_j^2\|^2}{1/\sigma_0^2 + 1/\sigma_j^2 + \sum_{-j,l} B_{i,l}^j/\sigma_j^2} - \frac{\|\sum_{-j,l} B_{i,l}^j \mathbf{v}_{jl}/\sigma_j^2\|^2}{1/\sigma_0^2 + \sum_{-j,l} B_{i,l}^j/\sigma_j^2} + \log \frac{m_i^{-j}}{J - m_i^{-j}}, & i \le L_{-j} \\ \frac{\|\mathbf{v}_{jl}/\sigma_j^2\|^2}{1/\sigma_0^2 + 1/\sigma_j^2} - 2\log(i - L_{-j}) + 2\log \frac{\gamma_0}{J}, & L_{-j} < i \le L_{-j}+L_j. \end{cases} \quad (9)$$

This completes the proof of Proposition 2 in the main text.

## 2 MULTILAYER INFERENCE DETAILS

Figure 1 illustrates the overall multilayer inference procedure visually, and Algorithm 1 provides the details.

---

**Algorithm 1** Multilayer Probabilistic Neural Matching

1: $L^{C+1} \leftarrow$ number of outputs
2: # Top down iteration through layers
3: **for** layers $c = C, C-1, \ldots, 2$ **do**
4:     Collect hidden layer $c$ from the $J$ servers and form $\mathbf{v}_{jl}^c$.
5:     Call Single Layer Neural Matching algorithm with output dimension $L^{c+1}$ and input dimension 0 (since we do not use the weights connecting to lower layers here).
6:     Form global neuron layer $c$ from output of the single layer matcher.
7:     $L^c \leftarrow \operatorname{card}(\cup_{j=1}^J \mathcal{T}_j^c)$ (greedy approach).
8: **end for**
9: # Match bottom layer using weights connecting to both the input and the layer above.
10: Call Single Layer Neural Matching algorithm with output dimension $L^2$ and input dimension equal to the number of inputs.
11: Return global assignments and form global mutltilayer model.

---

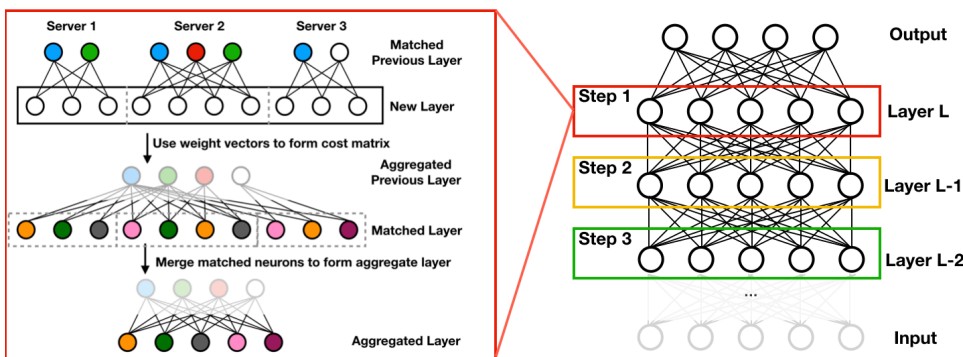

Figure 1: Probabilistic Neural Matching algorithm showing matching of three multilayer MLPs. Nodes in the graphs indicate neurons, neurons of the same color have been matched. On the left, the individual layer matching approach is shown, consisting of using the matching assignments of the next highest layer to convert the neurons in each of the $J$ servers to weight vectors referencing the global previous layer. These weight vectors are then used to form a cost matrix, which the Hungarian algorithm then uses to do the matching. Finally, the matched neurons are then aggregated and averaged to form the new layer of the global model. As shown on the right, in the multilayer setting the resulting global layer is then used to match the next lower layer, etc. until the bottom hidden layer is reached (Steps 1, 2, 3,... in order).

## 3 STREAMING NEURAL MATCHING

In this section we present inference for the streaming extension of our model described in Section 3.3 of the main text. Bayesian paradigm naturally fits into the streaming scenario - posterior of step $s$ becomes prior for step $s+1$:

$$\sigma_{0,i}^{2,s+1} = \frac{1}{1/\sigma_{0,i}^{2,s} + \sum_{j,l} B_{i,l}^{j,s}/\sigma_j^2}, \quad \boldsymbol{\mu}_{0,i}^{s+1} = \frac{\boldsymbol{\mu}_0^s/\sigma_{0,i}^{2,s} + \sum_{j,l} B_{i,l}^{j,s}\mathbf{v}_{jl}^s/\sigma_j^2}{1/\sigma_{0,i}^{2,s} + \sum_{j,l} B_{i,l}^{j,s}/\sigma_j^2} \text{ for } i = 1, \ldots, L^s. \quad (10)$$

The cost for finding $\boldsymbol{B}^{j,s}$ becomes:

$$\mathbf{C}_{i,l}^{j,s} = - \begin{cases} \frac{\|\boldsymbol{\mu}_{0,i}^s/\sigma_{0,i}^{2,s} + \mathbf{v}_{jl}^s/\sigma_j^2 + \sum_{-j,l} B_{i,l}^{j,s}\mathbf{v}_{jl}^s/\sigma_j^2\|^2}{1/\sigma_{0,i}^{2,s} + 1/\sigma_j^2 + \sum_{-j,l} B_{i,l}^{j,s}/\sigma_j^2} - \frac{\|\boldsymbol{\mu}_{0,i}^s/\sigma_{0,i}^{2,s} + \sum_{-j,l} B_{i,l}^{j,s}\mathbf{v}_{jl}^s/\sigma_j^2\|^2}{1/\sigma_{0,i}^{2,s} + \sum_{-j,l} B_{i,l}^{j,s}/\sigma_j^2} + \log \frac{1+m_{j,i}^s}{s-m_{j,i}^s}, \\ \frac{\|\boldsymbol{\mu}_{0,i}^s/\sigma_{0,i}^{2,s} + \mathbf{v}_{jl}^s/\sigma_j^2\|^2}{1/\sigma_{0,i}^{2,s} + 1/\sigma_j^2} - \frac{\|\boldsymbol{\mu}_{0,i}^s/\sigma_{0,i}^{2,s}\|^2}{1/\sigma_{0,i}^{2,s}} - 2\log(i - L_{-j}^s) + 2\log \frac{\gamma_0}{sJ}, \quad L_{-j}^s < i \leq L_{-j}^s + L_j^s, \end{cases} \quad (11)$$

where first case is for $i \leq L_{-j}^s$ and $m_{j,i}^s = \sum_{a=1}^{s-1}\sum_l B_{i,l}^{j,a}$ is the popularity of global atom $i$ in group $j$ up to step $s$. We note that $\log P(\boldsymbol{B}^{j,s}|\boldsymbol{B}^{-j,s}, \{\boldsymbol{B}^{j,1}\}_{j=1}^J, \ldots, \{\boldsymbol{B}^{j,s-1}\}_{j=1}^J)$ is not available in closed form in the Bayesian nonparametric literature, to the best of our knowledge, and we replaced corresponding terms in the cost with a heuristic.

## 4 EXPERIMENTAL DETAILS AND ADDITIONAL RESULTS

Code to reproduce our results will be released after the review period. Below are the details of the experiments.

**Data partitioning.**   In the federated learning setup, we analyze data from multiple sources, which we call batches. Data on the batches in general does not overlap and may have different distributions. To simulate federated learning scenario we consider two partition strategies of MNIST and CIFAR-10. For each pair of partition strategy and dataset we run 10 trials to obtain mean accuracies and standard deviations. The easier case is homogeneous partitioning, i.e. when class distributions on batches are approximately equal as well as batch sizes. To generate homogeneous partitioning with $J$ batches we split examples for each of the classes into $J$ approximately equal parts to form $J$ batches. In the heterogeneous case batches are allowed to have highly imbalanced class distributions as well as highly variable sizes. To simulate heterogeneous partition, for each class $k$, we sample $\mathbf{p}_k \sim \text{Dir}_J(0.2)$ and allocate $\mathbf{p}_{k,j}$ proportion of instances of class $k$ of the complete dataset to batch $j$. Note that due to small concentration parameter, 0.2, of the Dirichlet distribution, some batches may entirely miss examples of a subset of classes.

**Batch networks training.**   Our modeling framework and ensemble related methods operate on collection of weights of neural networks from all batches. Any optimization procedure and software can be used locally on batches for training neural networks. We used PyTorch (Paszke et al., 2017) as software framework and Adam optimizer (Kingma & Ba, 2014) with default parameters unless otherwise specified. For reproducibility we summarize all parameter settings in Table 1.

Table 1: Parameter settings for batch neural networks training

|  | MNIST | CIFAR-10 |
|---|---|---|
| Neurons per layer | 50 | 50 |
| Learning rate | 0.01 | 0.001 |
| $L_2$ regularization | $10^{-6}$ | $10^{-5}$ |
| Minibatch size | 32 | 32 |
| Epochs | 10 | 10 |
| Weights initialization | $\mathcal{N}(0, 0.01)$ | $\mathcal{N}(0, 0.01)$ |
| Bias initialization | 0.1 | 0.1 |

### 4.1 PARAMETER SETTINGS FOR THE BASELINES

We first formally define the ensemble procedure. Let $\hat{y}_j \in \Delta^{K-1}$ denote probability distribution over $K$ classes output by neural network trained on data from batch $j$ for some test input $x$. Then uniform ensemble prediction is $\arg\max_k \frac{1}{J} \sum_{j=1}^J \hat{y}_{j,k}$. To define weighted ensemble, let $n_{j,k}$ denote number of examples of class $k$ on batch $j$ and $n_k = \sum_{j=1}^J n_{j,k}$ denote total number of examples of class $k$ across all batches. Prediction of the weighted ensemble is as follows $\arg\max_k \frac{1}{n_k} \sum_{j=1}^J n_{j,k} \hat{y}_{j,k}$. This is a heuristic approach we defined to potentially better handle heterogeneous partitioning with ensemble.

Knowledge distillation approach (Hinton et al., 2015) trains a new master neural network to minimize cross entropy between output of the master neural network and outputs of the batch neural networks. The architecture of the master neural network has to be set manually - we use 500 neurons per layer. Note that PFNM infers the number of neurons per layer of the master network from the batch weights. For the knowledge distillation approach it is required to pool input data from all of the batches to the master server. For training master neural network we used PyTorch, Adam optimizer and parameter settings as in Table 1.

For the downpour SGD (Dean et al., 2012) we used PyTorch, Adam optimizer and parameter settings as in Table 1 for the local learners. Master neural network was also optimized with Adam and same

learning rate as in the Table 1. Weights of the master neural network were updated in the end of every epoch (total of 10 rounds of communication) and then sent to each of the local learners to proceed with the next epoch. Note that with this approach global network and networks for each of the batches are bounded to have identical number of neurons per layer, which is 50 in our experiments. We tried increasing number of neurons per layer, however did not observe any performance improvements.

## 4.2 ADDITIONAL EXPERIMENTAL RESULTS

**Master network size of PFNM.** Our model for matching neural networks is nonparametric and hence can infer appropriate size of the master network from the batch weights. The "discovery" of new neurons is controlled by the second case of our cost term expression in (9), i.e. when $L_{-j} < i \leq L_{-j} + L_j$. In practice however we want to avoid impractically large master networks, hence we truncate the largest possible value of $i$ in the cost computation to $\min(L_{-j} + L_j, \max(L_j, 700) + 1)$. This means that when global network has 700 or more neurons, we only allow for it to grow by 1 in a single multibatch Hungarian algorithm iteration.

In Figure 2 we summarize network sizes learned by PFNM in experiments corresponding to increasing number of batches (Figure 2 of the main text) and increasing number of hidden layers (Figure 3 of the main text). The maximum possible size is $50JC$ (because of 50 neurons per batch per layer), which is practically the size of the master model of the ensemble approaches. We see that size of the master network of PFNM is noticeably more compact than simply stacking batch neural networks. The saturation around 700 neurons in Figure 2a is due to the truncation procedure described previously.

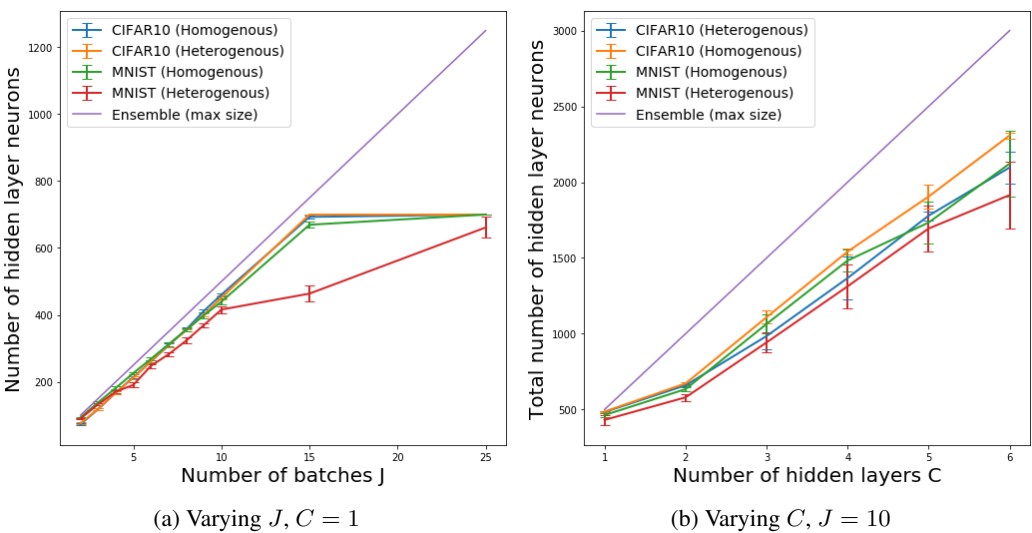

(a) Varying $J$, $C = 1$         (b) Varying $C$, $J = 10$

Figure 2: Network sizes for varying number of batches and layers

**Downpour SGD with frequent communication** In the main text we considered downpour SGD with total of 10 rounds of communication — one after each training epoch. This implies $20J$ communications, $10J$ for batches to send their gradients to the master server and $10J$ for the master server to send copy of the global neural network to each of the batch neural networks. In our federated learning problem setup frequent communication is discouraged, however it is interesting to study the minimum number of communications needed for D-SGD to produce competitive result. To empirically quantify this we show test accuracy of D-SGD with increasing number of communication rounds on MNIST with heterogeneous partitioning and $J = 25$ (Fig. 3a). PFNM and ensemble based methods are shown for comparison - they communicate only once (post batch networks training) in all of our experiments. We see that in this case D-SGD requires more than 4000 communication rounds ($8000J$ communications) to produce good result. Such large amount of communication is impossible in practice for the majority of federate learning scenarios.

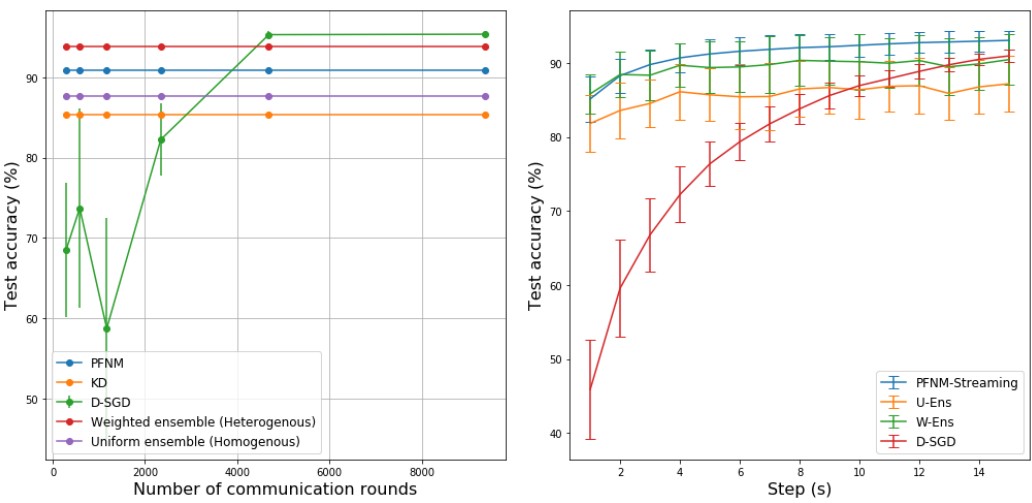

(a) D-SGD with varying number of communication rounds

(b) Streaming experiment on heterogeneous MNIST with $J = 3$, $S = 15$

Figure 3: Additional experiments

**Streaming federated learning experiment** To simulate streaming federated learning setup we partition data into $J$ groups using both homogeneous and heterogeneous strategies as before. Then each group is randomly split into $S$ parts. At step $s = 1, \ldots, S$ the part of data indexed by $s$ from each of the groups is revealed and used for updating the models. For this experiment we consider heterogeneous partitioning of MNIST with $J = 3$ batches and $S = 15$ steps. On every step we evaluate accuracy on the test dataset and summarize performance of all methods in Figure 3b. For our method, PFNM-Streaming, we perform matching based on the cost computations from Section 3 to update the global neural network. We initialize weights of the $j$-th batch neural network for the next step $s + 1$ according to the model posterior after $s$ steps (which is the prior for step $s + 1$): subsample $L_j^{s+1} - \gamma_0$ neurons from the global network according to popularity counts $\{m_{j,i}^s\}_i$ and concatenate with $\gamma_0$ neurons initialized with $\boldsymbol{\mu}_0^0$ (prior mean before any data is observed, which is set to 0), then add small amount of Gaussian noise. For D-SGD we update global neural network weights after each step and then use these weights as initialization for batch neural networks on the next step. To extend ensemble based methods to streaming setting we simply update local neural networks sequentially as the new data becomes available and use them to evaluate ensemble performance at each step.

## 4.3 PARAMETER SENSITIVITY ANALYSIS FOR PFNM

Our models presented in Section 3 of the main text have three parameters $\sigma_0^2, \gamma_0$ and $\sigma^2 = \sigma_1^2 = \ldots = \sigma_J^2$. The first parameter, $\sigma_0^2$, is the prior variance of weights of the global neural network. Second parameter, $\gamma_0$, controls discovery of new neurons and correspondingly increasing $\gamma_0$ increases the size of the learned master network. The third parameter, $\sigma^2$, is the variance of the local neural network weights around corresponding master network weights. We analyze empirically effect of these parameters on the accuracy for single hidden layer model with $J = 25$ batches in Figure 4. The heatmap indicates the accuracy on the training data - we see that for all parameter values considered performance doesn't not fluctuate significantly. PFNM appears to be robust to choices of $\sigma_0^2$ and $\gamma_0$, which we set to 10 and 1 respectively in all of our experiments. Parameter $\sigma^2$ has slightly higher impact on the performance and we set it using training data during experiments. To quantify importance of $\sigma^2$ for fixed $\sigma_0^2 = 10$ and $\gamma_0 = 1$ we plot average train data accuracies for varying $\sigma^2$ in Figure 5. We see that for homogeneous partitioning and one hidden layer $\sigma^2$ has almost no effect on the performance (Fig. 5a and Fig. 5c). In the case of heterogeneous partitioning (Fig. 5b and Fig. 5d), effect of $\sigma^2$ is more noticeable, however all considered values result in competitive performance.

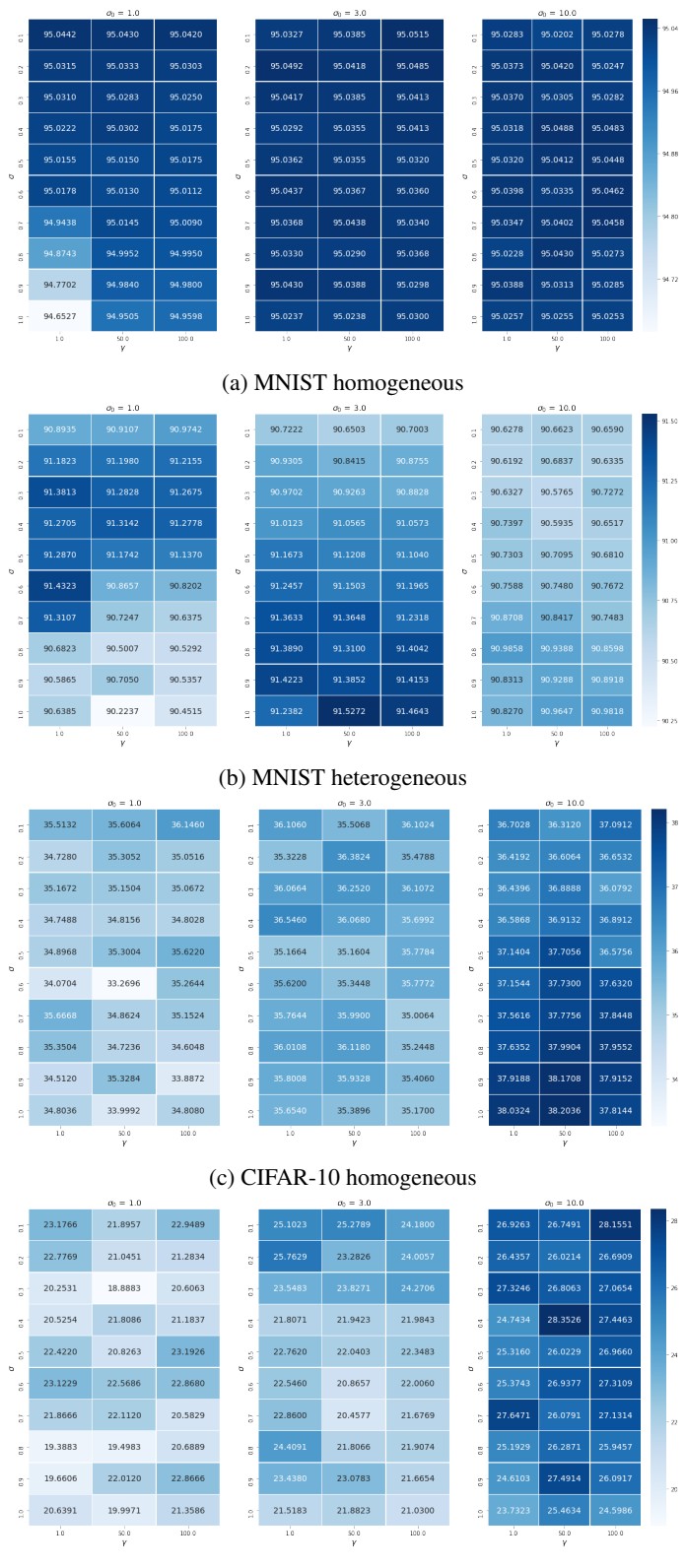

Figure 4: Parameter sensitivity analysis for $J = 25$

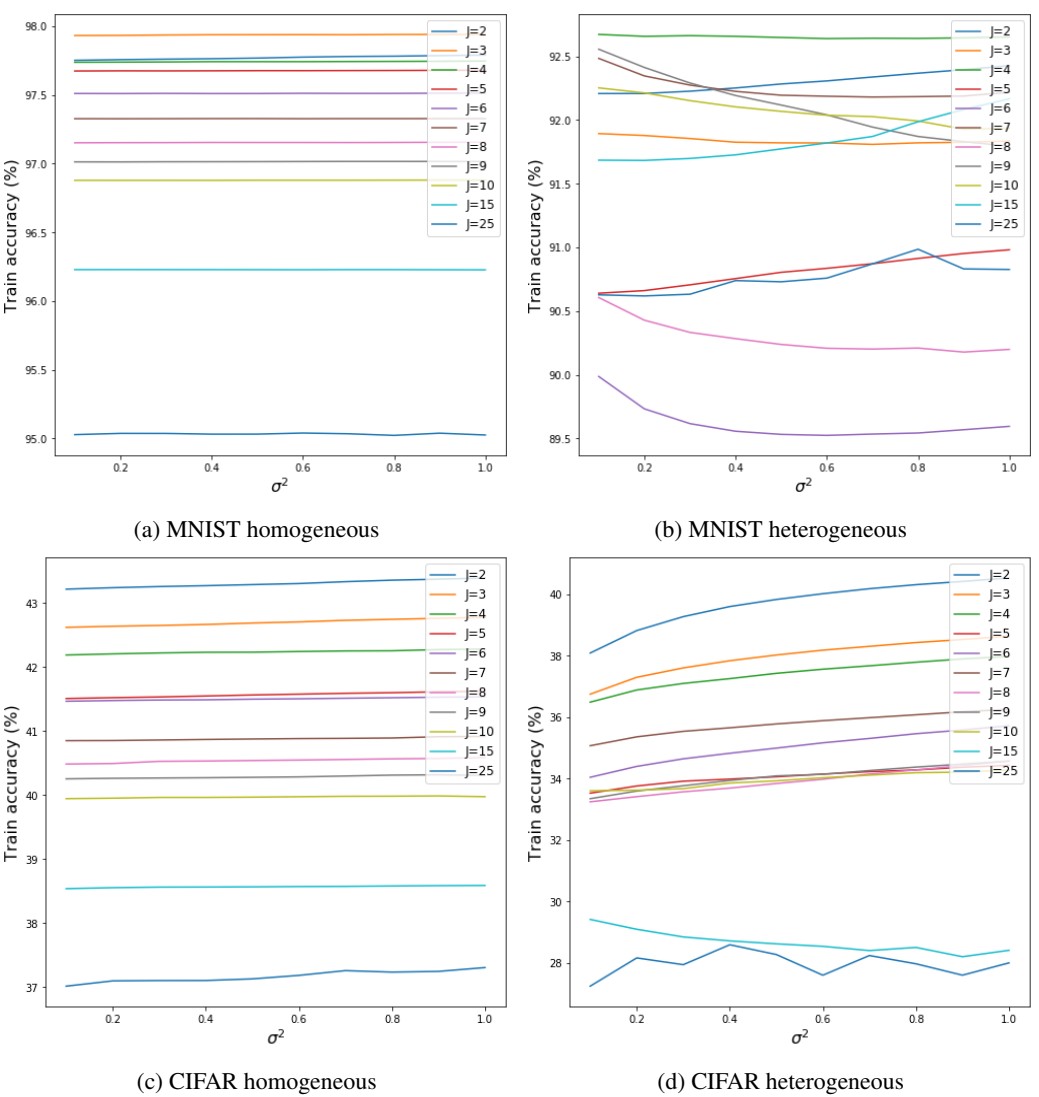

Figure 5: Sensitivity analysis of $\sigma^2$ for fixed $\sigma_0^2 = 10$ and $\gamma_0 = 1$ for varying $J$

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
