# OpenReview forum: "Probabilistic Federated Neural Matching"
_ICLR.cc/2019/Conference_

### Official Review · AnonReviewer3 · 2018-11-01
**interesting way to combine neural networks trained locally on federated data using a Beta-Bernoulli process**

**Rating:** 6
**Confidence:** 4

**Review:**

Summary: The paper considers federate learning of neural networks, i.e. data are distributed on multiple machines and the allocation of data points is potentially inhomogenous and unbalanced. The paper proposes a method to combine multiple networks trained locally on different data shards and form a global neural network. The key idea is to identify and reuse neurons that can be used for all shards and add new neurons to the global network if necessary. The matching and combination process is done via MAP inference of a model using a Beta-Bernoulli process. Some experiments on federated learning demonstrate the performance of the proposed method.

General evaluation (+ pro/ - con, more specific comments/questions below):
+ the paper is very well-written -- the BBP presentation is light but very accessible. The experimental set up seems sound.
+ the matching procedure is novel for federated training of neural networks, as far as I know, but might not be if you are a Bayesian nonparametric person, as the paper pointed out similar techniques have been used for topic models.
- the results seem to back up the claim that the proposed is a good candidate for combining networks at the end of training, but the performance is very similar or inferior to naive combination methods and that the global network is way larger than individual local network and nearly as large as simply aggregating all neurons together.
- the comparison to recent federated learning methods is lacking (e.g. McMahan et al, 2017) (perhaps less communication efficient than the proposed method, but more accurate).

Specific comments/questions/suggestions:
- the MAP update for the weights given the assignment matrix is interesting and resembles exactly how the Bayesian committee machine algorithm of Tresp (2000) works, except that the variances are not learnt for each parameter but fixed for each neuron. On this, there are several hyperparameters for the model, e.g. variance sigma_j -- how are these tuned/selected?
- the local neural networks are very mall (only 50 neurons per layer). How do they perform on the test set on the homogeneous case? Is there a performance loss by combining these networks together?
- the compression rate is not that fantastic, i.e. the global network tends to add new neuron for each local neuron considered. Is this because it is in general very hard to identify similar neuron and group them together? In the homogeneous case, surely there are some neurons that might be similar. Or is it because of the MAP inference procedure/local optima?

---

> ### Author Response · Authors · 2018-11-13
> **Authors response**
>
> We thank the reviewer for their time and interesting suggestions. We have added additional experiments to the draft (first paragraph of Section 4) to help address your concerns and we provide additional comments below.
>
> To address concerns regarding performance, compression rates, comparison to McMahan et al. (2017), and local neural networks we conducted an additional experiment (please see first paragraph of Section 4 and Fig. 2). In our previous experiments all `baselines' had some kind of intrinsic advantage over our setting and our goal was to achieve comparable performance, rather than outperform them. For example, improving upon an ensemble by averaging models in the parameter space seems very challenging -- if possible at all -- especially for higher number of batches and smaller batch sizes. A more fair comparison would be against approaches fully compatible with the federated learning problem studied in this work.
>
> In particular, we concur that the baselines you suggested are well suited as "fair" baselines, i.e. the performance of local neural networks and Federated Averaging of McMahan et al. (2017) with a single communication. McMahan et al. (2017)  (Figure 1 in their paper) empirically observed that sharing initialization across neural networks improves the performance of naive averaging of weights and used this idea to propose Federate Averaging. To implement Federated Averaging we shared initialization across batches and trained local neural networks with one hidden layer and 300 neurons for 10 epochs and then performed weighted averaging to obtain the global model. In Fig. 2 (main text) we see that even with shared initialization averaging of weights does not perform well for higher number of batches and/or heterogeneous data partitioning. In this experiment, for PFNM we trained local neural networks with 100 neurons each (and no constraints on how they were initialized). We considered two values of $\gamma_0$, one that results in larger model but better performance and a degenerate one resulting in the model of the size of the local net, but sacrificing performance quality. PFNM with $\gamma_0=1$ (truncated at 700 neurons; other hyperparameters were fixed to $\sigma^2_0=10$ and $\sigma^2_j=1$ for each j) consistently outperforms all baselines. In this experiment compression is relatively significant, i.e. when J=100, max size is 10000, whereas PFNM used around 500 neurons in the global model.
>
> We also note that when it is permissible to do multiple communication rounds to improve the performance, PFNM may serve as good initialization (setting $\gamma_0$ very small to enforce global model to be of the same size as local models) to Federated Averaging without the need to share common initialization across local neural networks.
>
> Regarding hyperparameters: when comparing to baselines satisfying the constraints (Fig. 2) we set $\sigma_j=1$ for every j and $\sigma^2_0=10$ across all experiments for fair comparison. When comparing to baselines with extra resources (Fig. 3) we set $\sigma^2_0=10$ and value of $\sigma_j$ was shared across $j$ and selected based on the train data performance. Please see section "Parameter sensitivity analysis for PFNM" in the Supplementary (Section 4.3, Fig. 4 and Fig. 5) for more details. In summary, we observed that $\sigma_j$ does not have much effect when partition is homogeneous and causes minor fluctuations in performance for heterogeneous cases.

---

### Official Review · AnonReviewer2 · 2018-11-03
**An interesting idea, but slightly contrived and lacking empirical support**

**Rating:** 6
**Confidence:** 4

**Review:**

The paper develops a novel solution for federated learning under three constraints, i.e. no data pooling (which distillation violates), infrequent communication (which iterative distributed learning violates), and modest-sized global model (which ensemble model violates). This is acknowledgedly a kind of unique setting, and the proposed solution does fit it well.

However, I have the following two main concerns
1. The major attack on distillation from an ensemble is that it needs to pool data across all sources which has cost and privacy concerns. However I'm not entirely convinced this "data pooling" is really necessary. One could argue distillation might as well be performed with simply an extra dataset that could be collected (sampled) elsewhere.
Plus, even though the proposed solution doesn't need to do "data pooling", it is effectively doing "model pooling" which may has its own costs and issues, e.g. the assumptions that one has access to all the parameters of the local models, and that all those local models should more or less be homogeneous to allow such pooling to happen, might not hold.

2. The idea of applying Beta-Bernoulli Process to uncover the underlying global model from a pool of local models is interesting. But I would very much like to see comparisons to some other simpler baselines, e.g. using dictionary learning to extract the common set of basis shared among the local models, or perhaps the slightly fancier DP-means (Kulis & Jordan, 2012)? Especially the lack of a meaningful improvement over the compared baselines from the empirical studies makes me wonder whether the BBP is indeed fit for purpose or even necessary for this task.

Some other questions/comments,
1. I'd be interested to see what the authors think about the connection between their proposed PFNM to Hinton's dropout, which could also be interpreted as performing an implicit "model pooling" over an ensemble of local models sharing weights among each other.

2. After introducing the notation for "-j", I'd suggest not to abuse "j" to keep denoting (dummy) indices in summations (e.g. Eq.(7), (8), etc.) - I might prefer swapping it with e.g. "j'" in $B^{j'}_{i,l}$, $v_{j'l}$ and $\sigma^2_{j'}$ to avoid confusions.

3. When the number of batches J gets larger, which means a smaller batch size and therefore also a larger variance among the local models, would it be beneficial to also increase the noise variances $\sigma_j$ accordingly to allow a better fit?

---

> ### Author Response · Authors · 2018-11-13
> **Authors response**
>
> We thank the reviewer for their time and interesting suggestions. We have added additional experiments to the draft (first paragraph of Section 4) to help address the concerns and we provide additional comments below.
>
> Regarding distillation from an ensemble and DP-means: We think that our method can be considered complementary to knowledge distillation when it is possible to obtain some amount of additional data. In particular, to train a distillation network, one would need to decide on the number of hidden neurons and the weight initialization for this network. Since local networks' weights are available, it is desirable to reuse them. However, a naive strategy of model averaging in the parameter space does not work well. McMahan et al. (2017) (see Figure 1 of their paper) empirically observed that sharing initialization across neural networks improves the performance of naive averaging of weights. DP-means, as you suggested, could be another option. It is also possible to simply pick one of the local models at random for initialization. In the added experiment (Fig. 2), we show that our PFNM method provides the best model pooling solution among all of these alternatives. These alternative methods are the more appropriate baselines for our method. In our previous experiments all `baselines' had some kind of intrinsic advantage over our setting and our goal was to achieve comparable performance, rather than outperform them. For example, improving upon an ensemble by pooling models in the parameter space seems very challenging, if at all possible, especially in the large number of batches and small batch size regime.
>
> Regarding dropout: The important difference between our setting and dropout (when viewed as model pooling) is that we aggregate networks trained independently, while dropout may be viewed as implicitly aggregating networks trained sequentially, i.e. each new network is initialized from a previous one. The permutation invariance phenomenon motivating PFNM implies that a neural network with L hidden neurons has at least $L!$ equivalent permuted neural networks that are equivalent local optima. When neural networks are trained independently, it is possible that they converge to similar, up to a permutation, solutions. That is one of the reasons naive averaging of weights (since it ignores permutations) performs poorly and matching based model pooling is more appropriate. The dropout case is the opposite of this, since when trained sequentially, it seems much less likely that a neural network will jump from one permutation invariant local optima to another, making naive averaging of weights obtained throughout training with dropout work well.
>
> Regarding choice of $\sigma_j$ for smaller batch sizes: Currently we set $\sigma_j$ to be same for all $j$. In Section 4.3, Fig. 4 and 5 of the Supplementary material we present some sensitivity analysis. Empirically $\sigma_j$ does not have much effect when partition is homogeneous and causes minor fluctuations in performance for heterogeneous cases. From the modeling perspective, higher $\sigma_j$ implies smaller global model size since local neurons assume higher variation and become "more willing" to be matched to existing global neurons.
>
> Thank you for the suggestions regarding notations - we will revise the manuscript accordingly.
>
> Reference: Brendan McMahan, Eider Moore, Daniel Ramage, Seth Hampson, and Blaise Aguera y Arcas.
> Communication-efficient learning of deep networks from decentralized data. In Artificial Intelligence and Statistics, pp. 1273–1282, 2017.

---

### Official Review · AnonReviewer1 · 2018-11-04
**Unclear advantage**

**Rating:** 4
**Confidence:** 3

**Review:**

The paper uses the beta process to do federated neural matching. The brief experimental results show worse performance than the other techniques compared with. Also, the motivation for the hierarchical beta process isn't clear, since each group has a single Bernoulli process. This makes learning each second level beta process a meaningless task. Why not have a single beta-Bernoulli process?

---

> ### Author Response · Authors · 2018-11-13
> **Authors response**
>
> We thank the reviewer for the feedback and provide answers to the raised concerns below.
>
> Regarding experiments: In our experiments each of the baselines is violating at least one of the constraints of the federated learning problem we are studying, i.e. single round of communication, no access to data after training local models and compressed global model. All baselines considered have a significant advantage by violating some of those conditions. Therefore the goal of the experiments was to show that we can achieve comparable performance with our method (PFNM) while adhering to all constraints, not to outperform the baselines. Indeed outperforming an ensemble by performing model averaging in the space of weights is extremely challenging, especially for many batches with fewer data points. We have added an additional experiment to compare with baselines satisfying all of the problem constraints. This experiment (Fig. 2) shows that PFNM outperforms all "fair" baselines by a good margin.
>
> Regarding Hierarchical Beta Process (HBP): We do not learn parameters of the second level Beta processes (except in the streaming case). Instead, those are integrated out and do not have any negative effect on the learning. We agree that it is possible to consider one global Beta process and a Bernoulli process per batch, however the group structure introduced by the second level Beta processes is important for the streaming case to infer heterogeneity of global atoms distributions across groups (Section 3.3; see Fig. 3b in Supplement for experimental evaluation of streaming case).

---

### Author Response · Authors · 2018-11-13
**Rebuttal**

We thank the reviewers for their feedback. We've uploaded the revised draft to resolve reviewers' concerns. Individual responses follow below.

---

### Meta-Review · Area_Chair1 · 2018-12-17
**Borderline paper**

**Confidence:** 4
**Recommendation:** Reject

**Metareview:**

While there was some support for the ideas presented, unfortunately this paper was on the borderline. Significant concerns were raised as to whether the setting studied was realistic, among others.